# Global characterization of gene expression in the brain of starved immature *Rhodnius prolixus*

**Jessica Coraiola Nevoa**[1], **Jose Manuel Latorre-Estivalis**[2], **Fabiano Sviatopolk-Mirsky Pais**[3], **Newmar Pinto Marliére**[1], **Gabriel da Rocha Fernandes**[3], **Marcelo Gustavo Lorenzo**[1], **Alessandra Aparecida Guarneri**[1] *

**1** Vector Behaviour and Pathogen Interaction Group, Instituto René Rachou – FIOCRUZ, Belo Horizonte, Minas Gerais, Brazil, **2** Laboratorio de Insectos Sociales, Instituto de Fisiología, Biología Molecular y Neurociencias, Universidad de Buenos Aires - CONICET, Buenos Aires, Argentina, **3** Bioinformatics Plataform, Instituto René Rachou - FIOCRUZ, Belo Horizonte, Minas Gerais, Brazil

* alessandra.guarneri@fiocruz.br

**Data Availability Statement:** The raw sequence dataset is available with the NCBI-SRA Bioproject number PRJNA853796 at NCBI.

## Abstract

### Background

*Rhodnius prolixus* is a vector of Chagas disease and has become a model organism to study physiology, behavior, and pathogen interaction. The publication of its genome allowed initiating a process of comparative characterization of the gene expression profiles of diverse organs exposed to varying conditions. Brain processes control the expression of behavior and, as such, mediate immediate adjustment to a changing environment, allowing organisms to maximize their chances to survive and reproduce. The expression of fundamental behavioral processes like feeding requires fine control in triatomines because they obtain their blood meals from potential predators. Therefore, the characterization of gene expression profiles of key components modulating behavior in brain processes, like those of neuropeptide precursors and their receptors, seems fundamental. Here we study global gene expression profiles in the brain of starved *R. prolixus* fifth instar nymphs by means of RNA sequencing (RNA-Seq).

### Results

The expression of neuromodulatory genes such as those of precursors of neuropeptides, neurohormones, and their receptors; as well as the enzymes involved in the biosynthesis and processing of neuropeptides and biogenic amines were fully characterized. Other important gene targets such as neurotransmitter receptors, nuclear receptors, clock genes, sensory receptors, and *takeouts* genes were identified and their gene expression analyzed.

### Conclusion

We propose that the set of neuromodulatory-related genes highly expressed in the brain of starved *R. prolixus* nymphs deserves functional characterization to allow the subsequent development of tools targeting them for bug control. As the brain is a complex structure that

**Funding:** AAG and MGL were supported by CNPq – Brazil productivity grants (AAG, grant number 303546/2018-2; MGL, grant number 311826/2019-9). This work was supported by Fundação de Amparo à Pesquisa do Estado de Minas Gerais (FAPEMIG, AAG, grant number CRA-PPM-00162-17; MGL, grant number APQ-01708-22), Instituto Nacional de Ciência e Tecnologia em Entomologia Molecular (INCTEM/CNPq, AAG, MGL, grant number 465678/2014-9), Conselho Nacional de Desenvolvimento Científico e Tecnológico (CNPq, MGL, grant number 308337/2015–8). This study was financed in part by the Coordenação de Aperfeiçoamento de Pessoal de Nível Superior – Brasil (CAPES) – Finance Code 001. The funders had no role in study design, data collection and analysis, decision to publish, or preparation of the manuscript.

**Competing interests:** The authors have declared that no competing interests exist.

presents functionally specialized areas, future studies should focus on characterizing gene expression profiles in target areas, e.g. mushroom bodies, to complement our current knowledge.

## Introduction

Triatomines are hematophagous insects that can transmit *Trypanosoma cruzi*, the etiological agent of Chagas disease. It is estimated that this neglected disease affects 7 million people, located mostly in Central and South America. The study of their biology is relevant because *T. cruzi* transmission is mostly controlled by eliminating domiciliated bugs [1].

Triatomines are nocturnal insects that assume an akinetic state while hidden in shelters during daylight hours. At nightfall, they eventually start a non-oriented locomotor activity, outside shelters to search for hosts. For host recognition, starved bugs detect cues released by vertebrates, such as radiant heat, water vapor, carbon dioxide, and other odorants [2]. The decision to leave a shelter and engage in foraging is risky, as triatomine hosts are often predators as well. For this reason, starved bugs mostly leave the protection of the shelters when a robust set of host clues is present [3–5]. Bugs of all nymphal instars and adults of both sexes feed on blood and can tolerate long starvation. Whereas nymphs have to feed to be able to molt, and adult females require nutrients to produce eggs [6]. Starved insects orientate towards host-emitted stimuli, while fed insects can remain indifferent or avoid these cues depending on the time elapsed after feeding [7].

The central nervous system (CNS) is the main regulator of physiology and behavior. Besides processing sensory information, the brain is the major accumulation of neuropiles integrating neural activity of sensory, memory, and proprioceptive nature [8]. As such, it has a main role in the coordination of motor responses, adjusting their proper timing through a set of clock neurons [9–11]. Signal transfer and modulation of neural processes in the CNS depend on neuroactive compounds, including neurotransmitters of diverse chemical nature like biogenic amines and neuropeptides, and their receptors [12, 13]. Neuropeptides and biogenic amines can also act as endocrine factors mediating signaling processes in multicellular organisms, and in the case of insects, they are fundamental in coordinating growth and development, as well as physiological processes such as metabolism, diuresis, digestion, reproduction, and behavior [14, 15]. *Rhodnius prolixus* Stål, 1859 (Hemiptera, Reduviidae, Triatominae) is considered an important vector of Chagas disease in Colombia and Venezuela due to its adaptability to both domestic and peridomestic environments, its rapid developmental cycle and the great population density it reaches in human dwellings [16–19]. Furthermore, *R. prolixus* has been widely used as a model for insect physiology studies, including research on reproduction, development, immunology, and vector-parasite interactions [20]. After the publication of its genome sequence [21], and the introduction of next-generation sequencing (NGS) methods, several studies have described genetic and molecular components underlying the physiology of *R. prolixus* [14, 15, 22–30]. Transcriptomic studies allowed the discovery of new genes and transcripts, the identification of differentially expressed genes, and determining targets for broader functional analyses. Some transcriptomes have analyzed gene expression in different *R. prolixus* tissues such as salivary glands [31], ovaries [32, 33], gut [29], testicles [34], and antennae [26]. Furthermore, this technique allowed defining the molecular bases of female reproductive physiology under differing nutritional states [22], as well as characterizing the innate immune system of these bugs at the molecular level [35]. Several of these bioinformatic analyses have recently shown that the genome of *R. prolixus* has many missing or miss-annotated genes [22,

33, 36], highlighting the importance of transcriptomes for improving the quality of the anno-tated genome of *R. prolixus*. Therefore, the present study aims to describe the genetic compo-nents that serve as the molecular neural bases controlling behavior in the brain of unfed *R. prolixus* nymphs.

## Materials and methods

### Insects

*Rhodnius prolixus* were obtained from a colony derived from insects collected in Honduras around 1990 and maintained by the Vector Behavior and Pathogen Interaction Group at the René Rachou Institute, Belo Horizonte, Brazil. Insects were monthly fed on citrated rabbit blood obtained from CECAL (Centro de Criação de Animais de Laboratório, FIOCRUZ, Rio de Janeiro, Brazil) offered through an artificial feeder at 37°C, alternating with feeding on anesthetized chicken and mice. Chickens were anesthetized with intraperitoneal injections of a mixture of ketamine (20 mg/kg; Cristália, Brazil) and detomidine (0.3 mg/kg; Syntec, Brazil), and mice with ketamine (150 mg/kg; Cristália, Brazil) and xylazine (10 mg/kg; Bayer, Brazil). Insects were reared in the insectary under $27 \pm 2°C$, $51 \pm 7\%$ of relative humidity and natural illumination. For this study, seven-day-old 4th instar nymphs were fed on citrated rabbit blood using an artificial feeder. Insects were kept unfed for 30 days after their ecdysis to the 5th instar and subsequently dissected to collect their brains. This study was carried out in strict accor-dance with the recommendations in CONCEA/MCT (http://www.cobea.org.br/), which is associated with the American Association for Animal Science (AAAS), the Federation of Euro-pean Laboratory Animal Science Associations (FELASA), the International Council for Ani-mal Science (ICLAS) and the Association for Assessment and Accreditation of Laboratory Animal Care International (AAALAC). The protocol was approved by the Committee for Eth-ics in the Use of Animals, CEUA, of the FIOCRUZ (Protocol Number: LW-61/2012).

### RNA extraction and illumina sequencing

Insect brains were dissected on a freeze cold dissecting dish (BioQuip, Gardena, CA, US), col-lected with forceps, and immediately transferred to a microtube immersed in dry ice and added 1 mL of TRIzol™ reagent (Invitrogen, Thermo Fisher Scientific, MA, USA). For sample completion, dissections occurred along three days, exclusively between 2 and 4 pm. Two sepa-rate experiments were performed, with three independent replicates each. Samples for repli-cates were, each composed of a pool of 20 brains. RNA extraction was performed with TRIzol™ according to the manufacturer's instructions. Total RNA concentrations were determined using a Qubit 2.0 Fluorometer (Life Technologies, Carlsbad, CA, US). Six libraries were con-structed using the TruSeq Stranded mRNA Sample Preparation Kit (Illumina, San Diego, CA) and sequenced on an Illumina HiSeq 2500 platform at the Max Planck Genome Center in Cologne (Germany). Approximately 15 million reads were obtained for each library, using 150 base-pair (bp) paired-end reads. The raw sequence dataset is available with the NCBI-SRA Bio-project number PRJNA853796 at NCBI.

### Bioinformatic analysis

Raw reads were filtered and trimmed for low-quality bases using Trimmomatic (v0.36) [37], according to standard quality score parameters (Phred-33 (>15); and 50 base-pair minimum length). Then, STAR v2.6.0 [38] was used with default parameters to map reads to the *R. pro-lixus* reference genome (version RproC3.3) accessed through the VectorBase website [39]. Mapped reads were assigned to each gene through the coverage tool in BEDTools (v2.29.2)

based on an updated gene annotation file [26]. A Principal Component Analysis (PCA) was performed through the *plotPCA* function in DESeq2 on RStudio to analyze the variation between the six libraries. Gene length and total counts of mapped reads were used for calculating Transcripts *per* Kilobase *per* Million mapped reads (TPM) values for target genes in each library. Subsequently, target gene expression (as $Log_{10}$ TPM+1) was depicted in heatmaps built using the pheatmap R package (v1.0.12). Finally, all genes were ranked according to the highest expression using TPM values, and the top 50 that were present in at least four of the six libraries, and which presented annotation in VectorBase [39] and/or a positive hit in BLASTp searches were selected. BLASTp searches were performed against Insecta class sequences in GenBank to identify putative functions of these highly expressed genes.

## Results and discussion

### Overall analysis

RNA-Seq data from starved $5^{th}$ instar *R. prolixus* brain transcriptomes were summarized in Table 1. After filtering and trimming raw reads, all libraries showed coverage of around 13 million reads. The number of uniquely mapped reads against the *R. prolixus* genome ranged from 8,8 M to 10,3 M reads. According to the PCA graph, three libraries clustered together (Rep2, Rep3 from Experiment 1 and Rep4 from Experiment 2), and three segregated apart (S1 Fig).

Based on our brain library outputs, six lists of the expressed genes were obtained after ranking their TPM values. To characterize the set of genes most highly expressed in the brain, we compared these ranks and built a consensus list depicting genes that ranked top 50 in at least four out of six libraries (Table 2). Several initiation and elongation factors together with ribosomal proteins were identified among top expressed genes, probably reflecting protein biosynthesis induced by starvation. Several heat shock proteins presented high expression in the brain, also probably due to starvation-generated stress. Different types of soluble carrier proteins, like odorant binding proteins, *takeouts* and lipocalins were very abundantly expressed in the CNS, suggesting roles other than odor transportation or detection. Finally, other highly expressed genes were the *neuroendocrine secretory protein 7B2* which functions as a specific chaperone for the prohormone convertase 2 (PC2) [40], an enzyme required for the maturation of neuropeptide and peptide hormone precursors; and the Glutamine synthetase that catalyzes the synthesis of glutamine, which has a central role in nitrogen metabolism and the regulation of neurotransmitter production [41].

### Neuropeptide precursor genes

Neuropeptide precursor gene (NPG) expression patterns in insects tend to be stereotyped, and each neuropeptide may be involved in neurotransmission, synaptic neuromodulation, or in

**Table 1. Summary of RNA-Seq metrics from R. prolixus brain transcriptomes.**

| Sample name | Raw reads | Clean reads | Uniquely mapped | Uniquely mapped (%) |
|:---:|:---:|:---:|:---:|:---:|
| **Brain_rep1** | 15,816,907 | 12,843,740 | 9,235,291 | 71.90 |
| **Brain_rep2** | 16,109,824 | 13,304,387 | 8,860,721 | 66.60 |
| **Brain_rep3** | 16,379,827 | 13,237,314 | 9,795,612 | 74.00 |
| **Brain_rep4** | 16,367,691 | 13,094,877 | 9,451,882 | 72.18 |
| **Brain_rep5** | 16,811,723 | 13,819,788 | 9,699,418 | 70.18 |
| **Brain_rep6** | 16,462,326 | 14,243,670 | 10,330,221 | 72.52 |

**Sample name:** name of replicate; **Raw Reads:** original sequencing reads; **Clean Reads:** number of reads after filtering; **Uniquely mapped:** number of reads that were uniquely mapped to the reference genome; **Uniquely mapped (%):** percentages of uniquely mapped reads.

**Table 2. Consensus list of the top 50 most highly expressed genes in brain transcriptomes of starved R. prolixus.**

| Gene ID | Gene Name | P length | Top hit of BLASTp against Insecta | Average TPM |
|---|---|---|---|---|
| RPRC006099 | NMDAr2a | 218 | Glutamate Receptor Ionotropic, Nmda 2b Isoform X2 [*Halyomorpha halys*] | 11244 |
| RPRC005193 | | 139 | Saga-Associated Factor 29 Homolog [*Fopius arisanus*] | 10131 |
| RPRC015041 | | 462 | Elongation Factor 1 Alpha [*Platymeris biguttatus*] | 10114 |
| RPRC000990 | | 702 | Hexamerin-Like [*Cimex lectularius*] | 9508 |
| RPRC004310 | | 652 | Heat Shock Cognate Protein [*Riptortus pedestris*] | 9005 |
| RPRC011668 | NPLP1 | 454 | Neuropeptide-Like Precursor 1 [*Rhodnius prolixus*] | 7909 |
| RPRC010283 | | 450 | Alpha-Tubulin 1 [*Lygus lineolaris*] | 7107 |
| RPRC009600 | | 302 | Mitochondrial Adp/Atp Translocase [*Triatoma infestans*] | 7020 |
| RPRC012247 | | 287 | Polyubiquitin-C Isoform X2 [*Pipistrellus kuhlii*] | 4930 |
| RPRC010096 | TO2 | 242 | Protein Takeout [*Cimex lectularius*] | 4589 |
| RPRC012142 | | 239 | Elongation Factor 2 [*Riptortus pedestris*] | 4506 |
| RPRC005793 | | 165 | Salivary Secreted Protein [*Triatoma infestans*] | 4120 |
| RPRC002589 | | 886 | Aminopeptidase N-Like [*Cimex lectularius*] | 3984 |
| RPRC009337 | | 274 | Ribosomal Protein S2 [*Riptortus pedestris*] | 3911 |
| RPRC009692 | | 141 | Secreted Hypothetical Protein [*Pristhesancus plagipennis*] | 3730 |
| RPRC005729 | CPR | 680 | Nadph Cytochrome P450 Reductase [*Triatoma infestans*] | 3623 |
| RPRC009568 | | 172 | Translationally Controlled Tumor Protein [*Riptortus pedestris*] | 3431 |
| RPRC012140 | | 605 | Elongation Factor 2 [*Riptortus pedestris*] | 3292 |
| RPRC011442 | | 179 | Nucleoplasmin-Like Protein Isoform X1 [*Cimex lectularius*] | 3179 |
| RPRC011742 | | 247 | 14-3-3 Protein Zeta Isoform X1 [*Cimex lectularius*] | 2996 |
| RPRC003327 | RpLP0 | 275 | 60s Acidic Ribosomal Protein P0 [*Halyomorpha halys*] | 2763 |
| RPRC009300 | | 320 | Polyadenylate-Binding Protein [*Riptortus pedestris*] | 2724 |
| RPRC017359 | pAbp1 | 628 | Polyadenylate-Binding Protein [*Riptortus pedestris*] | 2702 |
| RPRC012014 | | 249 | Protein Takeout-Like Isoform X1 [*Homalodisca vitripennis*] | 2670 |
| RPRC010786 | | 168 | Immunoglobulin Domain-Containing Protein [*Cimex lectularius*] | 2608 |
| RPRC009875 | | 376 | Actin-4 [*Bombyx mori*] | 2560 |
| RPRC015317 | | 356 | Arginine Kinase [*Triatoma infestans*] | 2468 |
| RPRC004408 | OBP11 | 128 | Heme-Binding Protein [*Rhodnius prolixus*] | 2186 |
| RPRC013825 | | 356 | Mitochondrial Phosphate Carrier Protein [*Riptortus pedestris*] | 2011 |
| RPRC001993 | | 192 | GTP-binding Protein REM 1-like Isoform X2 [*Cimex lectularius*] | 1859 |
| RPRC012101 | | 404 | S-Adenosylmethionine Synthase Isoform X2 [*Halyomorpha halys*] | 1840 |
| RPRC013341 | | 120 | Secreted hypothetical protein [*Pristhesancus plagipennis*] | 1827 |
| RPRC015183 | | 148 | Neuroendocrine Protein 7b2 Isoform X1 [*Halyomorpha halys*] | 1790 |
| RPRC007008 | OBP20 | 149 | Putative Odorant-Binding Protein [*Triatoma brasiliensis*] | 1784 |
| RPRC007612 | | 124 | Mite group 2 allergen Tyr p 2-like [*Cimex lectularius*] | 1766 |
| RPRC010050 | Tsf1 | 657 | Transferrin [*Rhodnius prolixus*] | 1735 |
| RPRC007515 | | 140 | Calmodulin Isoform X2 [*Nematostella vectensis*] | 1704 |
| RPRC000843 | Tachykinins | 215 | Tachykinin Precursor [*Rhodnius prolixus*] | 1691 |
| RPRC011264 | | 78 | Histone-lysine N-methyltransferase [*Nylanderia fulva*] | 1670 |
| ITG-like | ITG-like | 243 | Glutamine Synthetase Isoform X3 [*Halyomorpha halys*] | 1665 |
| RPRC001419 | | 745 | Venom periostin-like protein 1 [*Pristhesancus plagipennis*] | 1646 |
| RPRC000758 | | 243 | Glutamine Synthetase Isoform X3 [*Halyomorpha halys*] | 1636 |
| RPRC005040 | | 134 | Fatty Acid-binding Protein, muscle [*Cimex lectularius*] | 1632 |
| RPRC007684 | | 222 | Putative Elongation Factor 1 Beta [*Triatoma infestans*] | 1606 |
| RPRC012542 | RpS4 | 263 | 40S Ribosomal Protein S4 [*Cimex lectularius*] | 1577 |
| RPRC005701 | RpS8 | 208 | 40S Ribosomal Protein S8 [*Triatoma infestans*] | 1562 |
| RPRC007924 | | 323 | RNA-binding Protein Squid Isoform X5 [*Cimex lectularius*] | 1541 |

*(Continued)*

**Table 2.** (Continued)

| Gene ID | Gene Name | P length | Top hit of BLASTp against Insecta | Average TPM |
|---|---|---|---|---|
| RPRC004762 | Tubulin beta chain | 447 | Tubulin beta-1 chain isoform X2 [*Cimex lectularius*] | 1511 |
| RPRC006543 | ATPase 9 | 142 | ATP synthase lipid-binding protein [*Halyomorpha halys*] | 1491 |
| RPRC002700 | | 110 | Eukaryotic Translation Initiation Factor [*Halyomorpha halys*] | 1437 |

**Gene ID:** VectorBase gene code; **Gene name:** Gene name according to VectorBase; **P length:** Protein length; **Top hit of BLASTp against Insecta in GenBank:** Putative function of the gene according to BLASTp result; **Average:** Mean gene expression values among all six libraries in TPM.

conveying neuroendocrine signals at peripheral targets [12]. Currently, structural and functional studies on insect neuropeptides are being performed to develop new insect control approaches. This is especially true for neuropeptides involved in developmental, nutritional, and survival processes [13]. For this reason, we focused on the neuropeptides showing the highest expression in our study.

The four most highly expressed NPGs (Neuropeptide-like precursor 1—NPLP1, Tachykinin—TK, ITG-like, and NVP-like) presented expression values > 3 ($Log_{10}$ TPM+1). Most genes coding for neuropeptides (26 genes) had expression values between two and three, while eight had values between 1 and 2. A few of them, like ecdysis triggering hormone (*ETH*), elevenin 1 (*Ele1*), eclosion hormone (*EH*), sulphakinin (*SK*), and sifamide (*SIFa*) had expression values < 1. Adipokinetic hormone/corazonin-related peptide (*ACP*) was the only NPG out of 44 annotated for *R. prolixus* [26] that was not expressed in any of the six libraries here analyzed (Fig 1A).

Neuropeptide-like precursor 1 (NPLP1) was the NPG showing the highest expression in our transcriptomic database. Furthermore, it ranked amongst the 10 most highly expressed genes in the brain, presenting a mean of 3.90±0.03 $Log_{10}$ TPM+1 among our libraries. There were two different isoforms with distinct expression patterns previously detected in *R. prolixus*, isoform A expressed in CNS, ovaries, testes, and antennae, while expression of isoform B was only detected in antennae. The presence of mature peptides from NPLP1 precursors was also previously detected in the salivary glands of *R. prolixus* using proteomics [27]. The high expression observed for this neuropeptide precursor in our work suggests a relevant role in the CNS. Indeed, Sterkel et al. [30] showed that the levels of two mature peptides encoded by the *NPLP1* precursor gene significantly decreased in the CNS 24 hours after feeding, suggesting a role connected to starvation. A decrease in NPLP1 expression was also observed in the antennal lobe of mosquitoes after blood or sugar ingestion [42].

According to our brain dataset, TK was the NPG showing the second-highest expression level (3.22 ± 0.07 $Log_{10}$ TPM+1). Sterkel et al. [30] have also detected TK in the CNS of *R. prolixus* by means of peptidomics, not observing changes in its abundance after blood ingestion. This NPG presents expression in a wide set of *R. prolixus* tissues, as its transcripts have been detected in salivary glands, fat body, dorsal vessel, the intestinal tract of 5th instar nymphs (by RT-qPCR) [43], and in the antennae of 5th instar nymphs and adults (using RNA-Seq) [26]. Studies on other insects have shown that TKs can modulate early olfactory processing at the olfactory lobes, circuits controlling locomotion and food search, aggression, metabolic stress, and nociception [44]. Similar roles could be expected in triatomines based on the high expression of the *TK* gene observed in our study.

*ITG-like* and *NVP-like* genes also presented high expression values (3.21±0.08 and 3.10 ±0.04 $Log_{10}$ (TPM+1), respectively) in our database; nonetheless, to our knowledge, a functional characterization is not available for these peptides so far. Latorre-Estivalis et al. [26]

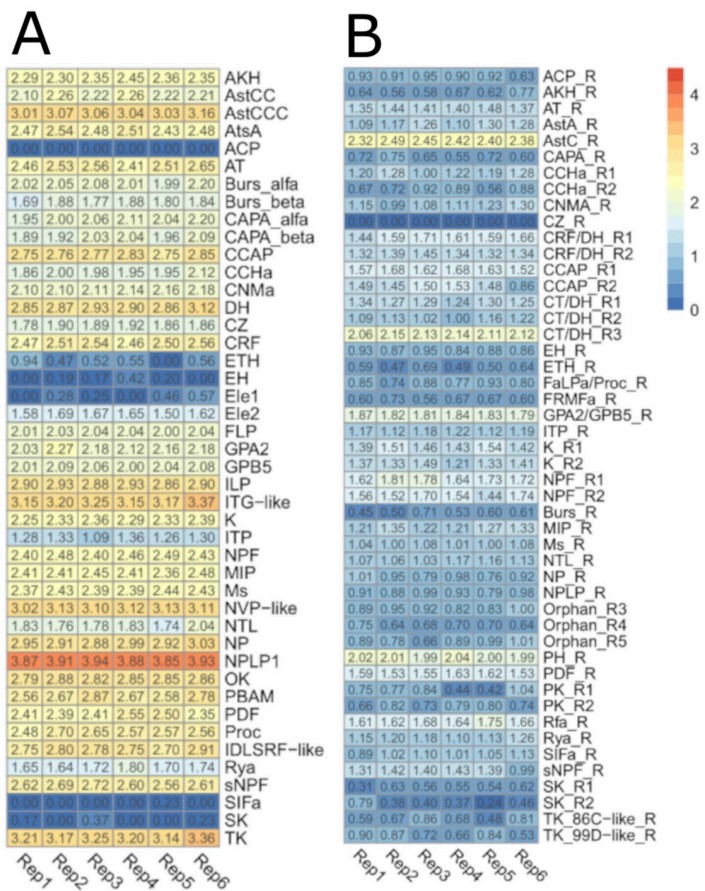

**Fig 1. Expression profiles of neuropeptide precursor and neuropeptide receptor genes.** (A) Heatmap depicting the expression level of neuropeptide precursor genes, and (B) neuropeptide receptor genes in the brain of *R. prolixus* nymphs. Expression (displayed as $Log_{10}$ TPM+1) is represented by means of a color scale in which blue/red represent the lowest/highest expression. Each column represents the expression of one library.

detected high expression of ITG-like in the antennae of immature and adults of *R. prolixus* and proposed a modulatory role for this neuropeptide at the peripheral level. In its turn, Leyria et al. [22] indicated that blood ingestion induced ovarian downregulation of the *NVP-like* gene, suggesting a role in *R. prolixus* reproduction [22]. The quantitative peptidomics analysis of the *R. prolixus* CNS by Sterkel et al. [30] showed that the abundance of ITG-like and NVP-like neuropeptides significantly decreases a few hours post-blood meal, indicating their implication in a neuroendocrine response to feeding. Considering this and the very high expression observed for ITG-like and NVP-like in our dataset from starved bugs, we suggest that they might act by signaling starvation status. Functional genetic studies based on gene silencing should be implemented to verify potential behavioral phenotypes that could offer evidence about the roles of these neuropeptides in *R. prolixus*.

Insulin-like peptide (*ILP*) presented a high expression in our database (2.90±0.02 $Log_{10}$ TPM+1). This result was consistent with the characteristics of our samples (brains from starved bugs) and the putative function of this neuropeptide as a modulator of lipid and carbohydrate metabolism [45, 46]. Indeed, *in vitro* immunofluorescence studies in *R. prolixus* brains demonstrated strong and abundant fluorescence of ILP neurons in unfed nymphs, followed by an acute decrease 4 hours after ingestion of a food meal, indicating transport and release of these signaling molecules into the hemolymph soon after feeding [47]. The high expression of

this neuropeptide in the CNS of unfed insects was also observed by Leyria et al. [22]. Interestingly, these authors observed that blood ingestion did not affect ILP gene expression in the CNS of *R. prolixus* [22].

## Neuropeptide processing enzymes

Mature neuropeptides are synthesized by a series of enzymatic steps that sequentially cleave and modify larger precursor molecules. One step of neuropeptide biosynthesis involves peptide amidation, a process that occurs on half of the known bioactive neuropeptides [48, 49]. The expression pattern observed for these enzymes might be a good proxy to estimate their abundance and activity [50], deserving their subsequent functional characterization.

A set of eleven neuropeptide processing enzymes was previously annotated for *R. prolixus* [26], using sequences of *D. melanogaster* as references [51]. All of these enzymes showed relative expression values higher than 1 ($\text{Log}_{10}$ TPM+1) (S2 Fig). Peptidyl alpha hydroxyglycine alpha amidatinglyase 2 (*PAL2*) and peptidylglycine α-amidating monooxygenase (*PHM*), the enzymes involved in amidation reactions, were the most highly expressed genes of this set (*PAL2*–2.92±0.07, *PHM*– 2.77±0.11).

## Neuropeptide receptors

Whether a tissue is targeted by a certain neuropeptide is defined by the presence of the corresponding neuropeptide receptor on the surface of its cells. Neuropeptide receptor genes showed much lower expression values than neuropeptide gene precursors. Only two out of the 48 neuropeptide receptor genes analyzed showed a relative expression higher than 2 (Fig 1B); allatostatin C (AstC) receptor with 2.38±0.05 $\text{Log}_{10}$ TPM+1 and calcitonin-like diuretic hormone receptor 3 -CT/DH-R3 with 2.12±0.03 $\text{Log}_{10}$ TPM+1. Twenty-six genes (54%) presented a relative expression between 2 and 1, and twenty genes (41%) had values lower than 1.

The expression of the AstC receptor in the *R. prolixus* CNS had been previously reported by Ons et al. [15] using RNA-Seq. These authors did not see expression changes for this receptor at the CNS after blood ingestion. Furthermore, Villalobos-Sambucaro et al. [52] observed the presence of this receptor in the hindgut, midgut and dorsal vessel, and showed that the receptor and its ligand play a key myoregulatory and cardioregulatory role in *R. prolixus*. Our results suggest that the AstC receptor may also have a fundamental role at the central level. Two receptors for CT/DH (named R1 and R2) were previously described in *R. prolixus*, their transcript expression being detected in the CNS and reproductive tissues [53]. The existence of a third CT/DH receptor in *R. prolixus* was suggested by Ons et al. [15] and confirmed by an antennal transcriptome [26]. The latter showed that this receptor had increased expression in male antennae when compared to those of nymphs, suggesting a stage-enriched role for this gene [26]. Our results suggest that this receptor may also act at the central level.

## Neurotransmitter receptors

As expected, neurotransmitter receptors tended to present higher expression values than the other receptor gene families studied here. N-methyl-D-aspartate receptor type 2A (*NMDAr2a*) was the second gene with the highest expression in the whole brain transcriptome (Fig 2; 4.05 ±0.03 $\text{Log}_{10}$ TPM+1). Muscarinic acetylcholine receptor type A (*AChR-A*) and dopamine 1-like receptor 1 (*DOP1*) were the other two genes that presented FPKM values higher than 2 in this gene set (2.16±0.12 $\text{Log}_{10}$ TPM+1 and 2.15±0.01 $\text{Log}_{10}$ TPM+1) for AChR-A and DOP1, respectively). Only two genes, 5-hydroxytryptamine (serotonin) receptors *1A* (*5HT-1A-R*) and *7A* (*5HT-7A-R*) showed values lower than 1 (0.62±0.08 and 0.82+0.08 $\text{Log}_{10}$ TPM +1) for *5HT-1A-R* and *5HT-7A-R*, respectively.

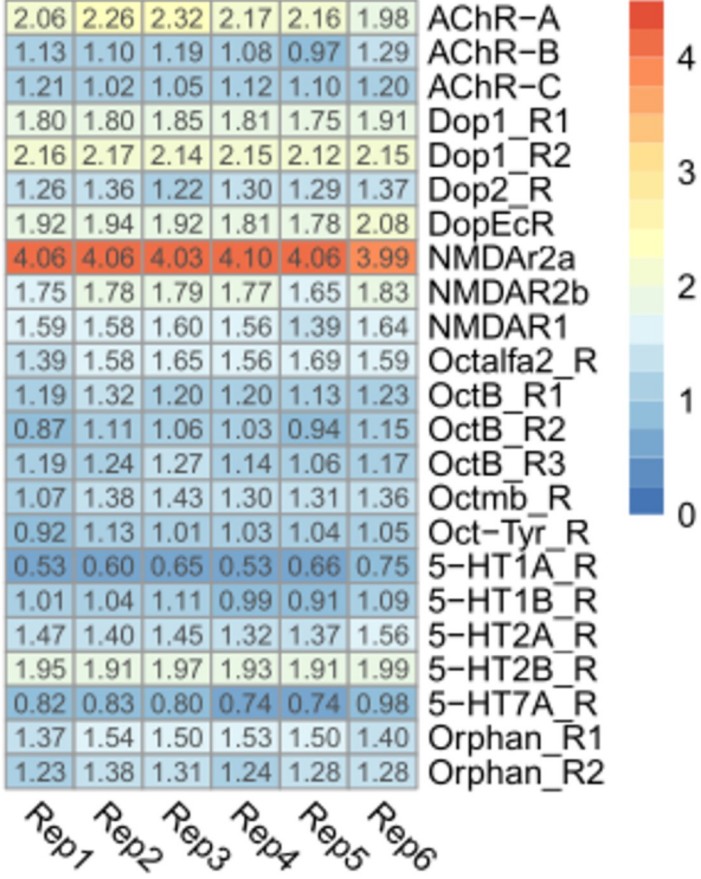

**Fig 2. Expression profiles of neurotransmitter receptors genes.** Heatmap depicting the expression level of neurotransmitter receptor genes in the brain of *R. prolixus* nymphs. Expression level (displayed as $Log_{10}$ TPM+1) is represented by means of a color scale in which blue/red represent the lowest/highest expression. Each column represents the expression of one library.

N-methyl-D-aspartate (NMDA) receptors are one of the subtypes of ionotropic receptors that bind to L-glutamate, mediating an excitatory activity in the CNS of insects. The NMDARs are usually constituted of two subunits NR1 and NR2 [54]. Even though these receptors were characterized in the brain of several invertebrate species, their functions in insects are poorly understood. However, their involvement in behavioral plasticity is already known [55, 56]. Similarly, a study of the evaluation of NMDAR expression in different tissues from female *Dactyola punctata*, showed that Dpun*NR1A*, Dpun*NR1B* and Dpun*NR2* were highly expressed in the brain [57]. In *Drosophila melanogaster*, both NMDA receptors called Dmel*NR1* and Dmel*NR2* were weakly expressed throughout the entire brain, with higher expression observed in some scattered cell bodies [58]. As far as we know, this is the first report on the expression of this receptor in the brain of *R. prolixus;* its high expression suggests a very relevant role in the neural physiology of these insects. Immunostaining experiments to characterize brain neuropiles depicting NMDAr2a expression will be required to initiate functional studies to uncover its putative function.

## Nuclear receptors

Most nuclear receptor genes presented low expression in the brain of unfed nymphs (Fig 3). Out of this gene set, the ecdysone-induced protein 75B (*Eip75B*) receptor was the only gene

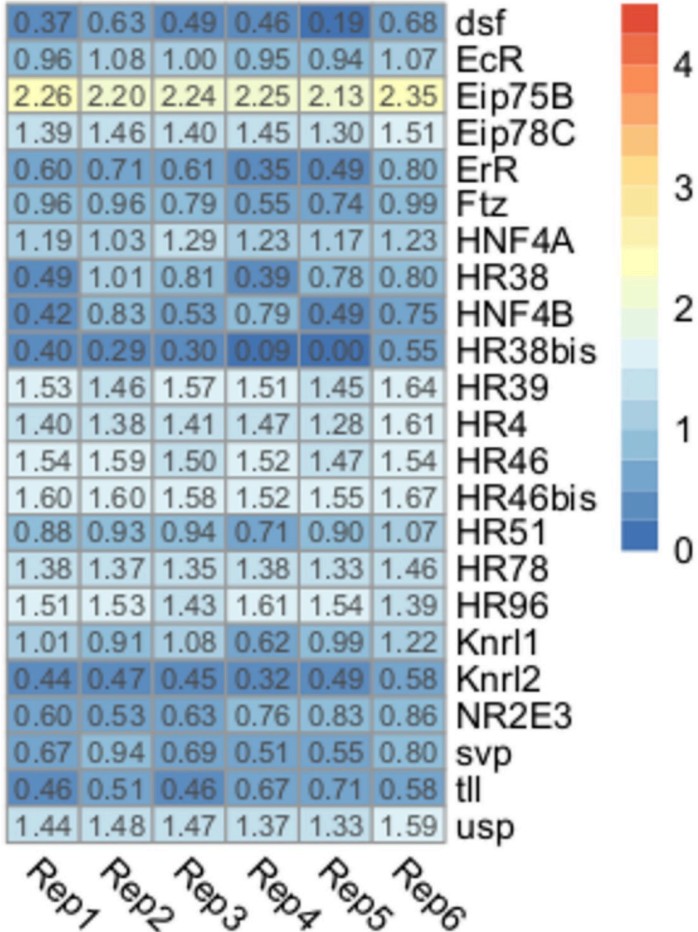

**Fig 3. Expression profiles of nuclear receptors genes.** Heatmap depicting the expression level of nuclear receptor genes in the brain of *R. prolixus* nymphs. Expression level (displayed as $Log_{10}$ TPM+1) is represented by means of a color scale in which blue/red represent the lowest/highest expression. Each column represents the expression of one library.

that presented TPM values higher than 2 (2.24±0.07 $Log_{10}$ TPM+1). Ten out of 23 genes (43%) presented values between 1 and 2 $Log_{10}$ TPM+1). The Eip75B and HR51 transcripts (the latter also known as *unfulfilled*) have been identified in central clock cells of *D. melanogaster* and control the expression of clock genes, playing an important role in the maintenance of locomotor rhythms [59–61]. Similar roles could be proposed for *R. prolixus*, however, functional information is not available for *Eip75B* and *HR51* genes in this species; only Latorre-Estivalis et al. [26] reported similar expression values for the *Eip75B* gene in kissing bug antennae.

## Clock and behavior-related genes

Clock genes are responsible for controlling circadian rhythms, and they can cycle in a synchronized way according to daily oscillations of environmental cues such as light and temperature [62]. Even though most available information on clock gene function has been generated using *D. melanogaster* as a model, their fundamental roles and their high level of sequence homology suggest that their functions should be conserved across insect orders. A total of 31 clock genes have been previously described and annotated in the *R. prolixus* genome [21]; however, as far

as we know, this is the first time that the expression of the whole clock gene set is studied in this insect. As all our brain samples were generated at the same interval of the daily cycle (2–4 PM), the expression profiles obtained here define the levels of expression of clock genes at this time. Except for *vrille* (*vri*), *cycle* (*cyc*), *single-minded* (*sim*) and *timeless* (*tim*) genes, the rest of the clock genes had expression higher than 1 $Log_{10}$ TPM+1 values (Fig 4). The most highly expressed genes (> 2 $Log_{10}$ TPM+1) were: casein kinase 2 (*ck2*), protein phosphatases (*Pp*) 1a and 2a, no circadian temperature entrainment (*nocte*), poly(A) binding proteins (*pAbp*) 1 and 2, and no receptor potential A (*norpA*). These genes showing high expression probably have key roles in the brain of unfed kissing bugs. Based on *D. melanogaster* studies, *pAbps* genes form a complex with *twenty-four* (*tyf*) and *Ataxin-2* (*Atx2*) that maintain circadian rhythms in

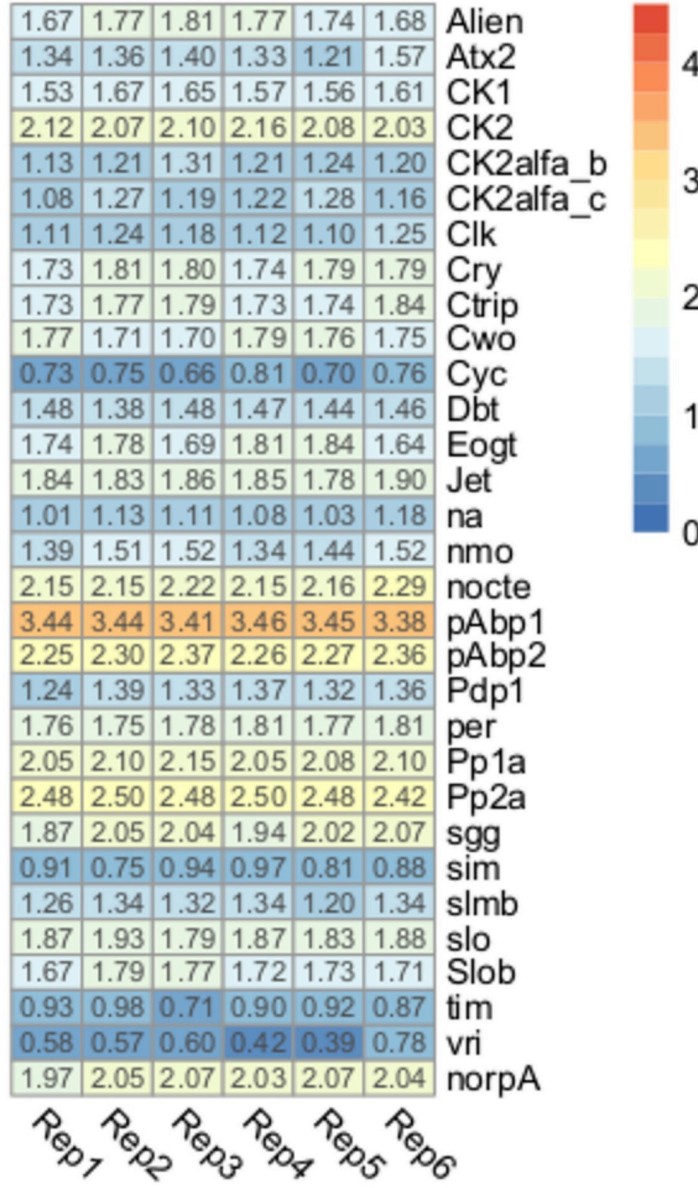

**Fig 4. Expression profiles of clock-related genes.** Heatmap depicting the expression level of clock-related genes in the brain of *R. prolixus* nymphs. Expression level (displayed as $Log_{10}$ TPM+1) is represented by means of a color scale in which blue/red represent the lowest/highest expression. Each column represents the expression of one library.

locomotor behavior [63]. The *Pp2A* gene, which is also highly expressed, controls the cyclic expression of the PER protein [64], which was also detected in our dataset. The *nocte* gene encodes a protein involved in temperature compensation of the circadian clock in *Drosophila*. It would be relevant to characterize 24h expression profiles of clock genes in the *R. prolixus* CNS to identify genes with cycling expression.

Other genes related to the control of insect behavior include *foraging* (*for*) whose expression was detected in our brain transcriptome (S1 Table). This gene encodes a cGMP-dependent protein kinase and plays an essential role in modulating food search in different species of insects, such as *D. melanogaster* [65–67], locusts [68], ants [69], honeybees [70], and social wasps [71]. In *R. prolixus*, *for* has been shown to participate in the modulation of locomotory activity [4, 72], and its expression in the brain and fat body changes depending on the nutritional status of the insect, increasing with starvation [72]. Our transcriptomic data seem to reinforce its relevance in the brain of *R. prolixus*.

### Sensory-related genes

Chemosensory proteins (CSPs) and odorant binding proteins (OBPs) can bind, solubilize and transport hydrophobic molecules [73]. The role of these transporters has been mainly studied in insect antennae and other sensory tissues where they bind odor molecules [73, 74].

Nevertheless, the expression of certain CSPs and OBPs has been reported in non-sensory tissues and involved in different functions, like releasing semichemicals in pheromone glands or associated to insecticide resistance, among others (revised by Pelosi et al. 2018) [75]. For this reason, we decided to data mine our database to characterize whether representatives of these protein families showed expression in bug brains (S2 Fig). Interestingly, high expression levels of several OBP and CSP transcripts, such as those of *RproCsp3*, *RproCsp5*, *RproCsp7*, *RproObp1*, *RproObp3*, *RproObp11*, *RproObp20*, and *RproObp26*, were detected in the brain of *R. prolixus*. The presence of these carrier proteins was previously reported in the brain of other insects [76–78]; however, its functional role in this tissue is still unknown. As proposed by Walker et al. (2019), these carriers could be monitoring internal chemical signals in the *R. prolixus* brain [76].

*Drosophila melanogaster takeout* 1 (*to*1) gene (*DmelTo1*) has been related to the regulation of feeding behavior and locomotor activity, and its expression has been detected in various fly structures and tissues, including the head, fat body, crop, and antennae [79, 80]. *DmelTo1* also affects male courtship behavior [81]. The role of these proteins has been poorly studied in insects other than Dipterans, even though to date the scarce evidence also points to behavioral roles in a locust and a moth [82, 83]. Regarding triatomines, the expression of TO genes has been reported in the digestive tract [29] and antennae of *R. prolixus* [26] and *T. brasiliensis* [84]. In our brain transcriptome, *RproTo1*, *RproTo2*, *RproTo4*, and *RproTo6* were highly expressed (S4 Fig), suggesting that they may have relevant behavioral functions at the central level, as observed for *D. melanogaster*.

### Final remarks

Expression datasets obtained using transcriptomes represent powerful tools to uncover molecular targets for functional studies. Furthermore, these data allow improving automatically predicted gene models of interest through the manual curation of sequences [21, 22]. Therefore, they also increase the chances of performing successful functional experiments based on more trustable gene models. A drawback associated with whole tissue transcriptomes is that complex structures like the brain can present an intricate organization with specialized areas having very differentiated functional roles, and consequently, specific gene expression profiles.

Therefore, brain transcriptome studies should acknowledge that expression profiles represent averages of neuropiles having differentiated properties. This is especially true for clock genes or NPGs which can be expressed in very restricted sets of neurons. Therefore, any lack of differential expression observed in studies comparing levels of expression in different developmental or physiological conditions should be later validated with tissue-specific or single-cell sequencing methods, when available. Still, the high levels of expression on which we decided to focus here seem to denote relevant functions that deserve attention, as they might guide research toward specific targets allowing the development of more rational control methods.

## Supporting information

**S1 Fig. Principal component analysis graph of the six libraries.** Rep 1, Rep 2, Rep 3 (experiment 1) and Rep 4, Rep 5, Rep 6 (experiment 2).
(TIF)

**S2 Fig. Expression profiles of neuropeptide processing enzymes genes.** Heatmap depicting the expression level of neuropeptide processing enzymes genes in the brain of *R. prolixus* nymphs. Expression level (displayed as $Log_{10}$ TPM+1) is represented by means of a color scale in which blue/red represent the lowest/highest expression. Each column represents the expression of one library.
(TIF)

**S3 Fig. Expression profiles of chemosensory proteins and odorant binding proteins.** (A) Heatmap depicting the expression level of chemosensory proteins (CSPs), and (B) odorant binding protein (OBPs) genes in the brain of *R. prolixus* nymphs. Expression level (displayed as $Log_{10}$ TPM+1) is represented by means of a color scale in which blue/red represent the lowest/highest expression. Each column represents the expression of one library.
(TIF)

**S4 Fig. Expression profiles of takeout genes.** Heatmap depicting the expression level of *takeout* genes in the brain of *R. prolixus* nymphs. Expression level (displayed as $Log_{10}$ TPM+1) is represented by means of a color scale in which blue/red represent the lowest/highest expression. Each column represents the expression of one library.
(TIF)

**S1 Table. Details of the mRNA expression of Figs 1–4, S2–S4 Figs.** Columns are: the abbreviation of the gene assigned; gene name according to the annotation; VectorBase code–the official gene number in the RproC3 genome assembly; values of TPM in each library (replicate); values of ($Log_{10}$ TPM+1) in each library. ND: not determined.
(XLSX)

## Author Contributions

**Conceptualization:** Jessica Coraiola Nevoa, Jose Manuel Latorre-Estivalis, Newmar Pinto Marliére, Marcelo Gustavo Lorenzo, Alessandra Aparecida Guarneri.

**Data curation:** Jessica Coraiola Nevoa, Jose Manuel Latorre-Estivalis, Fabiano Sviatopolk-Mirsky Pais, Gabriel da Rocha Fernandes.

**Formal analysis:** Jessica Coraiola Nevoa, Jose Manuel Latorre-Estivalis, Fabiano Sviatopolk-Mirsky Pais, Gabriel da Rocha Fernandes.

**Funding acquisition:** Marcelo Gustavo Lorenzo, Alessandra Aparecida Guarneri.

**Investigation:** Jessica Coraiola Nevoa, Jose Manuel Latorre-Estivalis, Alessandra Aparecida Guarneri.

**Methodology:** Jessica Coraiola Nevoa, Jose Manuel Latorre-Estivalis, Newmar Pinto Marliére, Alessandra Aparecida Guarneri.

**Project administration:** Alessandra Aparecida Guarneri.

**Resources:** Marcelo Gustavo Lorenzo, Alessandra Aparecida Guarneri.

**Software:** Jose Manuel Latorre-Estivalis, Fabiano Sviatopolk-Mirsky Pais, Gabriel da Rocha Fernandes.

**Supervision:** Marcelo Gustavo Lorenzo, Alessandra Aparecida Guarneri.

**Validation:** Jessica Coraiola Nevoa, Jose Manuel Latorre-Estivalis, Fabiano Sviatopolk-Mirsky Pais, Gabriel da Rocha Fernandes.

**Visualization:** Jessica Coraiola Nevoa, Fabiano Sviatopolk-Mirsky Pais, Gabriel da Rocha Fernandes, Marcelo Gustavo Lorenzo, Alessandra Aparecida Guarneri.

**Writing – original draft:** Jessica Coraiola Nevoa, Jose Manuel Latorre-Estivalis, Fabiano Sviatopolk-Mirsky Pais, Gabriel da Rocha Fernandes, Marcelo Gustavo Lorenzo, Alessandra Aparecida Guarneri.

**Writing – review & editing:** Jessica Coraiola Nevoa, Jose Manuel Latorre-Estivalis, Fabiano Sviatopolk-Mirsky Pais, Gabriel da Rocha Fernandes, Marcelo Gustavo Lorenzo, Alessandra Aparecida Guarneri.

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
