## [Decision Letter · Decision Letter 0]

1 Nov 2022

PONE-D-22-25346Global characterization of gene expression in the brain of starved immature Rhodnius prolixusPLOS ONE

Dear Dr. Guarneri,

Thank you for submitting your manuscript to PLOS ONE. After careful consideration, we feel that it has merit but does not fully meet PLOS ONE’s publication criteria as it currently stands. Therefore, we invite you to submit a revised version of the manuscript that addresses the points raised during the review process.  Although all of the Reviewer comments/suggestions are relevant and should be addressed to some degree, the most pertinent issues were: lack of methodological details for the starvation conditions, quality assessment of the replicated libraries and their overall utility for the analyses, suggested inclusion of GO terms, and the selection of Top 50 hits among four of the six libraries. In addition, I, like Reviewer 1, was a bit puzzled why fed Rhodnius were not included in the analyses. Given the reports in the literature re transcriptional regulation in response to blood feeding, it seems like directly comparing the two conditions would be more informative than one alone. I was also a bit surprised that AKH and sNPF transcript abundance was not higher. Could this be an artifact of the methodology or do you think it is real?  Line 152 should be changed to "around 13 million" rather "at least 13 million" as this is not supported by the data shown in Table 1 (ie rep 1 actually had less than 13 million reads).  Table 1 shows that 67-74% of the reads across the libraries were uniquely mapped. How does this value compare with other studies? 

We look forward to receiving your revised manuscript.

Kind regards,

J Joe Hull, Ph.D.

Academic Editor

PLOS ONE

Journal Requirements:

Reviewers' comments:

Reviewer's Responses to Questions

**Comments to the Author**

1. Is the manuscript technically sound, and do the data support the conclusions?

Reviewer #1: Partly

Reviewer #2: No

2. Has the statistical analysis been performed appropriately and rigorously? 

Reviewer #1: Yes

Reviewer #2: No

3. Have the authors made all data underlying the findings in their manuscript fully available?

Reviewer #1: Yes

Reviewer #2: No

4. Is the manuscript presented in an intelligible fashion and written in standard English?

Reviewer #1: Yes

Reviewer #2: Yes

5. Review Comments to the Author

Reviewer #1: In this study, Nevoa and collaborators describe transcriptome analyses in brain tissues of starved Rhodnius prolixus insects. Rhodnius belongs to the Triatomine family, which harbors hematophagous insect species associated with the transmission of the Chagas disease. Chags disease affects millions of people especially in Central and South America, but it still represents a clear example of neglected tropical diseases. By analyzing gene expression levels in the brain of starved fifth instar nymphs, the authors pinpoint several genes that might be involved in specific behaviors such as locomotion, aggression and food search among others. Although very descriptive, the manuscript is relevant and interesting especially because genomic and functional genomic studies in triatomines are still lagging behind. Such approaches might not only shed light on the biology of these insects, but also complete the annotation of the available genomes.

However, I have a few issues with the study. For instance, I do not understand why 6 replicates of starved nymphs were analyzed, instead of comparing fed versus starved animals. This would have been more informative about the gene expression profiles and how starving actually affects gene expression in the brain. Also, the choice of the stage is unclear to me. Why fifth instar nymphs instead of adults?

The authors only measure the expression in FPKM and then do, at least for me, an unusual selection of the top expressed genes (selection of top 50 in at least 4 out of 6?). From 100 top expressed genes in each library only 50 were common (sometimes not even in all replicates). I think it would be nice to have a quality assessment of the datasets, showing for instance how different libraries compare to each other (PCA or some distance clustering).

I believe that GO enrichment analysis should be performed. The way they do it throughout the paper is to get BLAST results from the selected genes and infer functions. GO analysis gives more statistical information about the different functions associated with a set of genes and probably should be done together with the BLAST analysis.

The quality of all the figures is below publication standards. I had hard times reading numbers and letters in all the heatmaps.

Line 137

"Gene length of mapped reads". The sentence should probably be "Gene length and total counts of mapped reads were used"

Line 138

The authors should use TPM instead of FPKM since FPKM (and RPKM) were already shown to be inconsistent across replicates (Wagner et al, 2012 - doi.org/10.1007/s12064-012-0162-3)

Line 143

"The identity and putative functions" should be changed to "To identify putative functions"

Reviewer #2: Several types of gene expression profiles in the brain of starved R. prolixus fifth instar nymphs were explored by means of RNA sequencing. However, I have some questions that the authors need to address. The quality of the manuscript could be improved. I regret that it may not meet the requirements for publication.

Major points:

Materials and methods

1. The aim of this study was to focus on the starved stages of R. prolixus but no description on how the nymphs were prepared.

2. Line 121-122: Six independent replicates was performed, however, it seems that no experiment designs were conducted, which made the results more descriptive but not problem-orientated.

3. The format of figures had a single form and the results seems to be descriptive and deductive.

Minor points:

Line 79: R. prolixus shall be more suitable to provide a full name for the first time. In addition, please add some explanation on the relationship between R. prolixus and triatomines.

Line 443-664: The format of reference should be carefully checked.

Table 2: Traces of revision was ignored.

6. PLOS authors have the option to publish the peer review history of their article (what does this mean?). If published, this will include your full peer review and any attached files.

Reviewer #1: No

Reviewer #2: No

---

## [Author Response · Author response to Decision Letter 0]

19 Dec 2022

Dear Dr J Joe Hull,

We are very grateful for the opportunity to respond to the reviewers’ comments and submit an improved version of our manuscript. Please, find below a point-by-point response to all comments. We hope that this new improved version can be considered for publication in PLOS ONE. 

Kind regards,

Alessandra Guarneri

In addition, I, like Reviewer 1, was a bit puzzled why fed Rhodnius were not included in the analyses. Given the reports in the literature re transcriptional regulation in response to blood feeding, it seems like directly comparing the two conditions would be more informative than one alone. 

Authors’ response: Regarding the use of six libraries of unfed bugs in our work, we want to clarify that they belong to an originally larger set of samples where the initial objective was to compare gene expression profiles between the brains of healthy and Trypanosoma cruzi-infected nymphs (infected either with the CL or Dm28c strains, each with 3 biological replicates). We decided not to include the comparison between infected and uninfected insects in this manuscript because the differential expression analyses showed no significant differences associated with parasite infection. Trypanosoma cruzi is a parasite that lives exclusively in the insect’s intestinal tract, and the changes in gene expression triggered in the R. prolixus brain may be subtle or restricted to few neurons in specific brain regions, being masked when the whole brain is analyzed. In fact, it has been shown that even the comparison between fed and unfed R. prolixus adults using RNA-Seq showed no significant changes in the expression of neuropeptide and neurohormonal signaling genes in the central nervous system (Leyria et al. 2020), reflecting the challenge of detecting gene expression differences in this tissue. Therefore, we decided to use the control samples (6 replicates) to describe the overall expression of genes in unfed nymphs because we felt that describing the expression of genes potentially involved in modulating behavior during starvation will contribute to a better understanding of the functional genomics of triatomines. Nevertheless, if the editor and reviewers consider it relevant to include the data referring to T. cruzi, we can prepare a new version of our manuscript containing the brain libraries of T. cruzi-infected nymphs.

Leyria J, Orchard I, Lange AB. Transcriptomic analysis of regulatory pathways involved in female reproductive physiology of Rhodnius prolixus under different nutritional states. Sci Rep. 2020;10(1).

I was also a bit surprised that AKH and sNPF transcript abundance was not higher. Could this be an artifact of the methodology or do you think it is real? 

Authors’ response: Following the editor’s comment, we have checked AKH and sNPF expression values in our databases and confirmed that both neuropeptides are expressed, presenting 835 and 918 raw counts; and 224.36 and 432.2 TPM, respectively (data are shown in S1_table). Considering that transcriptomes based on RNA samples of whole tissues are quite complex (cells are clearly not the same in expression profiles), it seems evident that these values should always be taken with a degree of caution before assuming the whole brain expresses these two genes with low expression. Indeed, it is known that the expression of many neuropeptide genes is restricted to groups of neurons. Therefore, a definitive answer will require new experiments tracking the presence of AKH and sNPF transcripts, e.g., in situ hybridization. Coincidently, RhoprSNPF peptides were detected in brain and hemolymph extracts by MS/MS (Ons et al., 2011), their presence in the CNS suggesting their local production, while hemolymph detection suggests a hormonal role (Ons, 2017).

Ons S. Neuropeptides in the regulation of Rhodnius prolixus physiology. J Insect Physiol. 2017;97: 77–92. 

Ons S, Sterkel M, Diambra L, Urlaub H, Rivera-Pomar R. Neuropeptide precursor gene discovery in the Chagas disease vector Rhodnius prolixus. Insect Mol Biol. 2011;20(1):29–44.

Line 152 should be changed to "around 13 million" rather "at least 13 million" as this is not supported by the data shown in Table 1 (ie rep 1 actually had less than 13 million reads). 

Authors’ response: We replaced “at least” by “around”, as requested. 

Table 1 shows that 67-74% of the reads across the libraries were uniquely mapped. How does this value compare with other studies? 

Authors’ response: The unique mapped read values obtained in our work are consistent with those obtained in other RNA-Seq studies performed in R. prolixus with the same methodology:

- Leyria, J., Orchard, I., & Lange, A. B. (2020). Transcriptomic analysis of regulatory pathways involved in female reproductive physiology of Rhodnius prolixus under different nutritional states. Scientific reports, 10(1), 1-16.

- Latorre-Estivalis, J. M., Große-Wilde, E., da Rocha Fernandes, G., Hansson, B. S., & Lorenzo, M. G. (2022). Changes in antennal gene expression underlying sensory system maturation in Rhodnius prolixus. Insect Biochemistry and Molecular Biology, 140, 103704.

- Coelho, V.L., de Brito, T.F., de Abreu Brito, I.A. et al. (2021). Analysis of ovarian transcriptomes reveals thousands of novel genes in the insect vector Rhodnius prolixus. Scientific Reports, 11, 1918.

Reviewer #1: 

However, I have a few issues with the study. For instance, I do not understand why 6 replicates of starved nymphs were analyzed, instead of comparing fed versus starved animals. This would have been more informative about the gene expression profiles and how starving actually affects gene expression in the brain. 

Authors’ response: We thank the reviewer for the interesting comments. Regarding the use of six libraries of unfed bugs in our work, we want to clarify that they belong to an originally larger set of samples where the initial objective was to compare gene expression profiles between the brains of healthy and Trypanosoma cruzi-infected nymphs (infected either with the CL or Dm28c T. cruzi strains, each with 3 biological replicates). We decided not to include the comparison between infected and uninfected insects in this manuscript because the differential expression analyses showed no significant differences associated with parasite infection. Trypanosoma cruzi is a parasite that lives exclusively in the insect’s intestinal tract, and the changes in gene expression triggered in the R. prolixus brain may be subtle or restricted to few neurons in specific brain regions, being masked when the whole brain is analyzed. In fact, it has been shown that even the comparison between fed and unfed R. prolixus adults using RNA-Seq showed no significant changes in the expression of neuropeptide and neurohormonal signaling genes in the central nervous system (Leryia et al. 2020), reflecting the challenge of detecting gene expression differences in this tissue. Therefore, we decided to use the control samples (6 replicates) to describe the overall expression of genes in unfed nymphs because we felt that describing the expression of genes potentially involved in modulating behavior during starvation will contribute to a better understanding of the functional genomics of triatomines. Nevertheless, if the reviewer considers it relevant to include the data referring to T. cruzi, we can prepare a new version of our manuscript containing the brain libraries of T. cruzi-infected nymphs.

Leyria J, Orchard I, Lange AB. Transcriptomic analysis of regulatory pathways involved in female reproductive physiology of Rhodnius prolixus under different nutritional states. Sci Rep. 2020;10(1).

Also, the choice of the stage is unclear to me. Why fifth instar nymphs instead of adults?

Authors’ response: We used 5th instar nymphs because we were interested in knowing the expression of genes involved in the modulation of behavior related to the search for blood sources, with no interference of those changes related to the search for a sexual partner (behavior that is exclusive of the adult phase).

The authors only measure the expression in FPKM and then do, at least for me, an unusual selection of the top expressed genes (selection of top 50 in at least 4 out of 6?). From 100 top expressed genes in each library only 50 were common (sometimes not even in all replicates). 

Authors’ response: We apologize if the choice for showing the 50th most expressed genes was not clear in the manuscript. The idea was to give to the reader a general idea about the genes with higher expression in the brain. The selection of this number is arbitrary, trying to avoid reporting a too extensive list. Therefore, we ranked library outputs according to gene FPKM values (now replaced by TPM), compared these ranks and built a consensus list depicting genes that ranked top 50 in at least four out of the six libraries. An additional criterion to complete this selection was that genes should present an annotation in VectorBase and/or a positive BLASTp output (against Insect class sequences in GenBanK) to grant informing their potential function. We have changed this information in the description of the results, that now reads:

“To characterize the set of genes most highly expressed in the brain, we compared these ranks and built a consensus list depicting genes that ranked top 50 in at least four out of six libraries”

I think it would be nice to have a quality assessment of the datasets, showing for instance how different libraries compare to each other (PCA or some distance clustering).

Authors’ response: Following the reviewer's suggestion, we added a PCA analysis in the new version of the manuscript. The S1 Fig includes a graph that summarizes the results of the PCA. Our results indicate that four libraries clustered together and two (one from each experiment) appeared segregated. To the best of our knowledge, we are satisfied with this result as it shows similar expression pattern throughout different experiments (see S1_table). Still, differences in gene expression where noticed and this could be related to the depth of the experiment itself.

I believe that GO enrichment analysis should be performed. The way they do it throughout the paper is to get BLAST results from the selected genes and infer functions. GO analysis gives more statistical information about the different functions associated with a set of genes and probably should be done together with the BLAST analysis.

Authors’ response: This is a very interesting suggestion. However, a proper GO enrichment analysis is based on the annotation of the GO-terms of a whole gene set. This process is currently undergoing work in R. prolixus. In fact, more than 30% of the R. prolixus genes has no single GO annotation (information from VectorBase). We consider that using such a limited background to identify GO enriched terms could lead to false conclusions regarding the R. prolixus gene dataset. Therefore, we would like to rely on our results based on BLAST hits as originally presented.

The quality of all the figures is below publication standards. I had hard times reading numbers and letters in all the heatmaps.

Authors’ response: We apologize for this. We have included new figures with the quality requested in the PLOS instructions for authors.

Line 137 "Gene length of mapped reads". The sentence should probably be "Gene length and total counts of mapped reads were used"

Authors’ response: The change was made as suggested.

Line 138 The authors should use TPM instead of FPKM since FPKM (and RPKM) were already shown to be inconsistent across replicates (Wagner et al, 2012 - doi.org/10.1007/s12064-012-0162-3)

Authors’ response: We appreciate the recommendation. Therefore, we have made new calculations to express all the values as TPM. 

Line 143 "The identity and putative functions" should be changed to "To identify putative functions"

Authors’ response: The change was made as requested.

Reviewer #2: 

Major points: Materials and methods

The aim of this study was to focus on the starved stages of R. prolixus but no description on how the nymphs were prepared.

Authors’ response: We apologize for the mistake. The following sentence was added to the text: 

“For this study, seven day-old 4th instar nymphs were fed on citrated rabbit blood using an artificial feeder. Insects were kept unfed for 30 days after their ecdysis to the 5th instar and subsequently dissected to collect their brains.”

Line 121-122: Six independent replicates was performed, however, it seems that no experiment designs were conducted, which made the results more descriptive but not problem-orientated.

Authors’ response: We thank the reviewer for the interesting comments. Regarding the use of six libraries of unfed bugs in our work, we want to clarify that they belong to an originally larger set of samples where the initial objective was to compare gene expression profiles between the brains of healthy and Trypanosoma cruzi-infected nymphs (infected either with the CL or Dm28c T. cruzi strains, each with 3 biological replicates). We decided not to include the comparison between infected and uninfected insects in this manuscript because the differential expression analyses showed no significant differences associated with parasite infection. Trypanosoma cruzi is a parasite that lives exclusively in the insect’s intestinal tract, and the changes in gene expression triggered in the R. prolixus brain may be subtle or restricted to few neurons in specific brain regions, being masked when the whole brain is analyzed. In fact, it has been shown that even the comparison between fed and unfed R. prolixus adults using RNA-Seq showed no significant changes in the expression of neuropeptide and neurohormonal signaling genes in the central nervous system (Leryia et al. 2020), reflecting the challenge of detecting gene expression differences in this tissue. Therefore, we decided to use the control samples (6 replicates) to describe the overall expression of genes in unfed nymphs because we felt that describing the expression of genes potentially involved in modulating behavior during starvation will contribute to a better understanding of the functional genomics of triatomines. Nevertheless, if the reviewer considers it relevant to include the data referring to T. cruzi, we can prepare a new version of our manuscript containing the brain libraries of T. cruzi-infected nymphs.

Leyria J, Orchard I, Lange AB. Transcriptomic analysis of regulatory pathways involved in female reproductive physiology of Rhodnius prolixus under different nutritional states. Sci Rep. 2020;10(1).

The format of figures had a single form and the results seems to be descriptive and deductive.

Authors’ response: The figures try to show, using different colors, the differences in the level of expression of genes belonging to several families. We have standardized them to allow the reader to have some degree of comparison.

Minor points:

Line 79: R. prolixus shall be more suitable to provide a full name for the first time. In addition, please add some explanation on the relationship between R. prolixus and triatomines.

Authors’ response: We apologize for the mistake. We included the full name of R. prolixus, as well as its biological classification.

Line 443-664: The format of reference should be carefully checked.

Authors’ response: The format of the references has been checked as requested.

Table 2: Traces of revision was ignored.

Authors’ response: We have checked and corrected the forgotten revision traces.

---

## [Decision Letter · Decision Letter 1]

30 Jan 2023

PONE-D-22-25346R1Global characterization of gene expression in the brain of starved immature Rhodnius prolixusPLOS ONE

Dear Dr. Guarneri,

Thank you for submitting your manuscript to PLOS ONE. After careful consideration, we feel that it has merit but does not fully meet PLOS ONE’s publication criteria as it currently stands. Therefore, we invite you to submit a revised version of the manuscript that addresses the points raised during the review process, in particular comments regarding the inclusion of BUSCO analyses and the potential role of Trypanosoma cruzi in the analyses.

We look forward to receiving your revised manuscript.

Kind regards,

J Joe Hull, Ph.D.

Academic Editor

PLOS ONE

Journal Requirements:

Reviewers' comments:

Reviewer's Responses to Questions

**Comments to the Author**

1. If the authors have adequately addressed your comments raised in a previous round of review and you feel that this manuscript is now acceptable for publication, you may indicate that here to bypass the “Comments to the Author” section, enter your conflict of interest statement in the “Confidential to Editor” section, and submit your "Accept" recommendation.

Reviewer #2: All comments have been addressed

Reviewer #3: (No Response)

2. Is the manuscript technically sound, and do the data support the conclusions?

Reviewer #2: Partly

Reviewer #3: Yes

3. Has the statistical analysis been performed appropriately and rigorously? 

Reviewer #2: Yes

Reviewer #3: N/A

4. Have the authors made all data underlying the findings in their manuscript fully available?

Reviewer #2: No

Reviewer #3: Yes

5. Is the manuscript presented in an intelligible fashion and written in standard English?

Reviewer #2: Yes

Reviewer #3: Yes

6. Review Comments to the Author

Reviewer #2: For the minor points, I think that the authors have fully addressed my points.

However, the major question on sample collection should be further issued with possible revision.

The authors stated thoroughly on initial objective of the manuscript which was in line with my speculation. I agreed with the Reviewer 1 that the samples of fed nymphs should be included in comparative transcriptome analysis to provide valuable information on the scientific question that the authors raised. Although Trypanosoma cruzi lives exclusively in the insect’s intestinal tract, it can elicit an immune response in Rhodnius prolixus as a parasite when brain may be involved. In this way, is there any possible interaction between starvation and brain response with or without infection? Starvation treatment may block the infection-related brain response which might be an interesting question. I suggest the authors to add some analysis on these points.

Reviewer #3: The authors report a descriptive analysis of global gene expression in brain of Rhodnius prolixus nymphs. Data is derived from 6 replicates, and analyses are conducted on the 50 most highly expressed genes in 4 of 6 samples, as well as several different gene families known to function in brain physiology. This version of the manuscript has been previously reviewed and revised, and the authors have addressed most of the authors concerns. The manuscript is thus mostly suitable for publication in current form, however, some improvements are suggested, mainly related to providing additional context to the results in the “Results and Discussion” section. Examples of this are provided below.

In response to the previous reviewer comments, the authors have included a PCA analysis of their datasets. It would also be informative to conduct BUSCO analyses on the transcriptomes derived from the genome mapping results. This would give a better indication of the completeness of each data set, and could be done in a relatively straightforward manner through the gVolate server (https://gvolante.riken.jp/)

Line 152-153. “According to the PCA graph, four libraries clustered together and two segregated apart.” In looking at this supplemental figure, it is not clear which libraries clustered together and which segregated apart. As it is shown, it appears that 3 are clustering together and 3 apart. So it would be useful to draw a circle around the four that are clustering together. Furthermore, mention is made to samples D, E, F, J, K, L, in the figure legend. However it is not clear what these designations refer to. In Table 1, and throughout the remainder of the figures, all of the samples are referred to as “Rep1, Rep2, Rep3, Rep4, Rep5, Rep6.” It would be very helpful to clarify somewhere what these different designations correspond to in Figure S1.

Line 197-199. “As expected for some of them…had expression values less than 1”. Why was this outcome expected? It would be good to clarify here why this qualifier was made? Do these genes not have function in nymphs in general, or brain tissue specifically?

Line 267-269. “The brain expression pattern shown by this group of genes seems to differ when compared to that shown in an antennal transcriptome” When this is mentioned, it must be clarified how they differ? Some context here would be appropriate.

Line 289-291. “The latter showed that this receptor had increased expression in the male antennae when compared to those of nymphs, suggesting a sexually dimorphic role for this peptide” How does this comparison suggest a sexually dimorphic role? Wouldn’t that be indicated by increased expression in male antennae compared to those of females? As written now, it sounds like what is suggested is stage-enriched role for the peptide, not sexually dimorphic.

Line 310-311. “The NMDARs are usually constituted of two subunits NR1 and NR2” Were both of these subunits found in the brain here? Looking at Figure 2, I see NMDAr2a (very highly expressed) and NMDAr2b, but no mention indication of any NMDAr1? Since it is mentioned about the two subunits, it would be good to clarify the findings in R. prolixus here.

Line 383-384. “Nevertheless, the expression of certain CSPs and OBPs has been reported for insect guts, testes, Malpighian tubules and salivary glands” It is also noted that a previous report on a moth (Spodoptera littoralis) showed expression of CSPs and OBPs in male and female adult brain tissue. See Walker et al., 2019 BMC Genomics. However, I am uncertain if expression of these gene families have been examined in the brain of any other hemipteran species. Furthermore, it may be useful to the reader to provide some comments on any potential roles for these carrier proteins in brain physiology.

7. PLOS authors have the option to publish the peer review history of their article (what does this mean?). If published, this will include your full peer review and any attached files.

Reviewer #2: No

Reviewer #3: **Yes: **William Walker

---

## [Author Response · Author response to Decision Letter 1]

14 Feb 2023

Dear Dr J Joe Hull,

Thank you for providing us with the comments from peer reviewers. We are glad for the opportunity to strengthen this manuscript. We have addressed all the reviewers’ points and remain excited for the prospect of our work to be featured in PLOS One.

Kind regards,

Alessandra Guarneri

Reviewer #2: 

The authors stated thoroughly on initial objective of the manuscript which was in line with my speculation. I agreed with the Reviewer 1 that the samples of fed nymphs should be included in comparative transcriptome analysis to provide valuable information on the scientific question that the authors raised. Although Trypanosoma cruzi lives exclusively in the insect’s intestinal tract, it can elicit an immune response in Rhodnius prolixus as a parasite when brain may be involved. In this way, is there any possible interaction between starvation and brain response with or without infection? Starvation treatment may block the infection-related brain response which might be an interesting question. I suggest the authors to add some analysis on these points.

Authors’ response: We thank the reviewer for the comments and agree that the comparison between fed and unfed insects would have improved the manuscript. Indeed, it is possible that some alterations in the brain of infected insects could have appeared if fed. Nevertheless, as we did not sequence samples of fed insects, and we are not including results from brains of infected insects in our manuscript, we understand that we cannot even speculate on whether starvation could impair the gene expression changes triggered by T. cruzi infection. 

Reviewer #3: 

1. In response to the previous reviewer comments, the authors have included a PCA analysis of their datasets. It would also be informative to conduct BUSCO analyses on the transcriptomes derived from the genome mapping results. This would give a better indication of the completeness of each data set, and could be done in a relatively straightforward manner through the gVolate server (https://gvolante.riken.jp/)

Authors’ response: We thank the reviewer for this constructive input, even though we understand it does not apply to our case. As the aim of our study was to quantify gene expression in the brain of starved bugs, we used the reference genome from VectorBase (a curated genome) for mapping purposes. As we understand that BUSCO would search for housekeeping genes to infer the completeness of an assembled transcriptome, we consider that it would not be adequate as we did not generate an assembly based on our transcriptome reads. Information about the reference genome used can be found at https://vectorbase.org/vectorbase/app/record/dataset/DS_b8c0427e28

2. Line 152-153. “According to the PCA graph, four libraries clustered together and two segregated apart.” In looking at this supplemental figure, it is not clear which libraries clustered together and which segregated apart. As it is shown, it appears that 3 are clustering together and 3 apart. So it would be useful to draw a circle around the four that are clustering together. Furthermore, mention is made to samples D, E, F, J, K, L, in the figure legend. However it is not clear what these designations refer to. In Table 1, and throughout the remainder of the figures, all of the samples are referred to as “Rep1, Rep2, Rep3, Rep4, Rep5, Rep6.” It would be very helpful to clarify somewhere what these different designations correspond to in Figure S1.

Authors’ response: We thank the reviewer for this helpful comment. We totally agree with the observation that it appears that only 3 libraries clustered together, so we corrected the phrase in line 155-157 that now reads:

“According to the PCA graph, three libraries clustered together (Rep2, Rep3 from Experiment 1 and Rep4 from Experiment 2), and three segregated apart (S1 Fig).”

We also have made changes in the PCA figure, now referring to the samples as “Rep1, Rep2, Rep3, Rep4, Rep5, Rep6.” 

3. Line 197-199. “As expected for some of them…had expression values less than 1”. Why was this outcome expected? It would be good to clarify here why this qualifier was made? Do these genes not have function in nymphs in general, or brain tissue specifically?

Authors’ response: We appreciate this comment because it made us perceive that this statement was not correct. We have modified the phrase that now reads as “A few of them, like ecdysis triggering hormone (ETH), elevenin 1 (Ele1) (...)”

4. Line 267-269. “The brain expression pattern shown by this group of genes seems to differ when compared to that shown in an antennal transcriptome” When this is mentioned, it must be clarified how they differ? Some context here would be appropriate.

Authors’ response: Again, we would like to thank the reviewer for this constructive input. Given that the functional information regarding several of these genes is rather limited, we removed this phrase to avoid offering an ambiguous perspective. 

5. Line 289-291. “The latter showed that this receptor had increased expression in the male antennae when compared to those of nymphs, suggesting a sexually dimorphic role for this peptide” How does this comparison suggest a sexually dimorphic role? Wouldn’t that be indicated by increased expression in male antennae compared to those of females? As written now, it sounds like what is suggested is stage-enriched role for the peptide, not sexually dimorphic.

Authors’ response: We thank the reviewer for this relevant comment. Indeed, the antennal expression of this gene was increased in the antennae of adults compared to nymphs. Consequently, we have modified the phrase that now reads “The latter showed that this receptor had increased expression in adult antennae when compared to those of nymphs, suggesting a stage-enriched role for this gene.”

6. Line 310-311. “The NMDARs are usually constituted of two subunits NR1 and NR2” Were both of these subunits found in the brain here? Looking at Figure 2, I see NMDAr2a (very highly expressed) and NMDAr2b, but no mention indication of any NMDAr1? Since it is mentioned about the two subunits, it would be good to clarify the findings in R. prolixus here.

Authors’ response:: We would like to thank the reviewer for this comment that made us perceive that we missed inclusion of the NNMDaR1 gene in figure 2. The new version of the manuscript includes the expression data for NNMDaR1 in Figure 2.

7. Line 383-384. “Nevertheless, the expression of certain CSPs and OBPs has been reported for insect guts, testes, Malpighian tubules and salivary glands” It is also noted that a previous report on a moth (Spodoptera littoralis) showed expression of CSPs and OBPs in male and female adult brain tissue. See Walker et al., 2019 BMC Genomics. However, I am uncertain if expression of these gene families have been examined in the brain of any other hemipteran species. Furthermore, it may be useful to the reader to provide some comments on any potential roles for these carrier proteins in brain physiology.

Authors’ response: We thank the reviewer for this comment and for providing an appropriate reference to improve our manuscript. After revising the literature for this topic, we would like to mention that no brain/head transcriptome seems to have been performed with hemipterans in which the presence of OBPs and CSPs were reported. Besides the suggested reference to WalKer et al. 2019 (BMC Genomics), we have found other papers that identified expression of OBPs and CSPs in the brain/head of several insects using RNA-Seq and qPCR. Therefore, we have edited this part of the manuscript including all these references and adding a comment on the potential roles of OBPs and CSPs in the brain of R. prolixus.

The new paragraph included reads as: “Nevertheless, the expression of certain CSPs and OBPs has been reported in non-sensory tissues and suggested to be involved in different functions, like releasing semiochemicals in pheromone glands or mediating insecticide resistance, among others (revised by Pelosi et al. 2018) (75). For this reason, we data mined our database to characterize whether representatives of these protein families were expressed in bug brains (S2 Fig). Interestingly, high expression levels of several OBP and CSP transcripts, such as those of RproCsp3, RproCsp5, RproCsp7, RproObp1, RproObp3, RproObp11, RproObp20, and RproObp26, were detected in the brain of R. prolixus. The expression of carrier protein coding genes, like those of OBPs and CSPs, was previously reported in the brain of other insects (76-78); however, their functional roles in this tissue are still unknown. As proposed by Walker et al. (2019), these carrier proteins may monitor internal chemical signals in the brain (76).”

---

## [Editor Report · Decision Letter 2]

16 Feb 2023

Global characterization of gene expression in the brain of starved immature Rhodnius prolixus

PONE-D-22-25346R2

Dear Dr. Guarneri,

We’re pleased to inform you that your manuscript has been judged scientifically suitable for publication and will be formally accepted for publication once it meets all outstanding technical requirements.

Kind regards,

J Joe Hull, Ph.D.

Academic Editor

PLOS ONE
---

## [Editor Report · Acceptance letter]

22 Feb 2023

PONE-D-22-25346R2 

Global characterization of gene expression in the brain of starved immature Rhodnius prolixus 

Dear Dr. Guarneri:

I'm pleased to inform you that your manuscript has been deemed suitable for publication in PLOS ONE. Congratulations! Your manuscript is now with our production department. 

Kind regards, 

on behalf of

Dr. J Joe Hull 

Academic Editor

PLOS ONE